# NegoCollab: A Common Representation Negotiation Approach for Heterogeneous Collaborative Perception

**Congzhang Shao**[1] **Quan Yuan**[1*] **Guiyang Luo**[1*] **Yue Hu**[2] **Danni Wang**[1]
**Yilin Liu**[1] **Rui Pan**[1] **Bo Chen**[1] **Jinglin Li** [1]

[1]State Key Laboratory of Networking and Switching Technology,
Beijing University of Posts and Telecommunications
[2]Cooperative Medianet Innovation Center,
Shanghai Jiaotong University
{shaocongzhang,yuanquan,luoguiyang}@bupt.edu.cn

## Abstract

Collaborative perception improves task performance by expanding the perception range through information sharing among agents. Immutable heterogeneity poses a significant challenge in collaborative perception, as participating agents may employ different and fixed perception models. This leads to domain gaps in the intermediate features shared among agents, consequently degrading collaborative performance. Aligning the features of all agents to a common representation can eliminate domain gaps with low training cost. However, in existing methods, the common representation is designated as the representation of a specific agent, making it difficult for agents with significant domain discrepancies from this specific agent to achieve proper alignment. This paper proposes NegoCollab, a heterogeneous collaboration method based on the negotiated common representation. It introduces a negotiator during training to derive the common representation from the local representations of each modality's agent, effectively reducing the inherent domain gap with the various local representations. In NegoCollab, the mutual transformation of features between the local representation space and the common representation space is achieved by a pair of sender and receiver. To better align local representations to the common representation containing multimodal information, we introduce structural alignment loss and pragmatic alignment loss in addition to the distribution alignment loss to supervise the training. This enables the knowledge in the common representation to be fully distilled into the sender. The experimental results demonstrate that NegoCollab significantly outperforms existing methods in common representation-based collaboration approaches. The mechanism of obtaining common representations through negotiation provides a more reliable and flexible option for common representations in heterogeneous collaborative perception.

## 1 Introduction

Collaborative perception has gained significant attention in recent years. By sharing intermediate features among agents, it expands the perception range and provides more supporting information for downstream tasks. In autonomous driving, collaborative perception enables vehicles to detect obstacles in blind spots, thereby preventing traffic accidents and effectively enhancing driving safety. Heterogeneity is one of the key challenges in practical applications of collaborative perception

---

*Corresponding author

39th Conference on Neural Information Processing Systems (NeurIPS 2025).

Xu et al. (2023b); Lu et al. (2024); Gao et al. (2025). When there are differences in sensors and perception models among collaborating agents, it creates domain gaps in the shared intermediate features. This prevents effective fusion of features from heterogeneous agents and consequently degrades collaborative performance.

Current research on heterogeneity issues includes approaches that achieve heterogeneous collaboration by retraining specialized collaborative modules Xiang et al. (2023) or sharing partial networks in model Lu et al. (2024). However, in practical deployment, perception model are crucial for autonomous driving safety and tightly coupled with downstream tasks, making it difficult to replace or retrain. These limitations lead to the challenge of **immutable heterogeneous collaborative perception** Xia et al. (2024). To address this issue, methods like Xu et al. (2023b); Luo et al.; Xia et al. (2024) employ domain adapters or polymorphic prompts to eliminate domain gaps through one-to-one adaptation for heterogeneous agents, as is shown in Figure 1a, requiring only single-step feature transformation but incurring higher training costs. Alternatively, Gao et al. (2025) aligns the representations of each modality's agent to a common representation by training a pair of adapter and reverter, which has low training cost. However, since the common representation is designated as the representation of a specific agent, as is shown in Figure 1b, alignment becomes difficult to achieve when there exists a large domain gap among the representations of other agents and the designated agent.

This paper presents NegoCollab, a heterogeneous collaborative framework based on negotiated common representation. The framework introduces an additional negotiator during training to generate common representation from local representations of each modality's agent, as is shown in Figure 1c, supervised by a cyclic distribution consistency loss. This design minimizes information loss during bidirectional transformation between local representations and the common representation, effectively reducing inherent domain discrepancies between them. During collaboration, NegoCollab facilitates heterogeneous information exchange through a pair of plug-and-play sender-receiver. The sender first maps features to the common representation space for sharing with collaborators, while the receiver subsequently projects the received features back to the local representation space, thereby eliminating domain gaps with collaborators' features. Furthermore, to better align local representations with the common representation containing multimodal information, structural alignment loss and pragmatic alignment loss are introduced in addition to the commonly used distribution alignment loss. These losses collectively form a multi-dimensional alignment loss to supervise the training, enabling the knowledge in the common representation to be fully distilled into the sender.

The main contributions of this work are summarized as follows:

- Introducing a negotiator to generate the common representation from local representations of each modality's agent, effectively reducing the alignment difficulty between the local representations and common representation while providing more diverse and reliable options for the common representations required in heterogeneous collaborative perception.

- A multi-dimensional alignment loss comprising distribution alignment loss, structural alignment loss, and pragmatic alignment loss is introduced to supervise the training process, enabling more effective alignment of local representations to the multimodal common representation.

- Experimental results on collaborative perception datasets demonstrates that NegoCollab achieves state-of-the-art performance among common representation-based methods, outperforming even one-to-one adaptation approaches in certain collaborative scenarios.

## 2 Related Work

### 2.1 Collaborative Perception

In recent years, collaborative perception has attracted widespread attention due to its potential to enhance autonomous driving safety. By sharing perception data among agents—including raw sensor data Rauch et al. (2012); Luo et al. (2023); Liu et al. (2024), intermediate features Wang et al. (2020); Li et al. (2021); Chen et al. (2019); Hu et al. (2022), and detection results Xu et al. (2023a); Rawashdeh and Wang (2018), collaborative perception effectively expands the perception range and overcomes

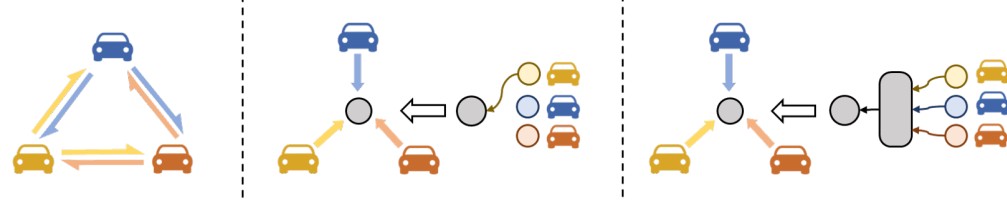

(a) One-to-one Adaption      (b) Align to Common (Designated)      (c) Align to Common (Negotiated)

Figure 1: Two paradigms for eliminating domain gaps. The method in (a) eliminates the domain gap by adapting domain adaptation modules between every pair of collaborating agents. The methods in (b) and (c) both eliminate domain gaps by unifying the representations of each agent into the common representation, where the common representation in (b) is designated as the local representation of a specific agent, and the common representation in (c) is negotiated from the local representations of each modality's agent.

blind spots and occlusion issues inherent in single-agent perception. However, in real-world scenarios, collaborative perception faces multiple challenges including: limited communication bandwidth Hu et al. (2022, 2023, 2024), location noise Lu et al. (2023); Lei et al. (2024), communication delay and computation asynchronously Lei et al. (2022); Wei et al. (2024a), communication interruptions Ren et al. (2024), heterogeneity Xu et al. (2023b); Xiang et al. (2023); Lu et al. (2024); Luo et al.; Gao et al. (2025); Xia et al. (2024), security and privacy concerns Li et al. (2023); Zhao et al. (2023), and simulation-to-real generalization issues Kong et al. (2023); Wei et al. (2024b), all of which pose challenges to collaboration. This paper focuses on the heterogeneity challenge in collaborative perception, proposing a negotiated common representation-based approach to achieve common representation-based heterogeneous collaboration.

## 2.2 Multi-modal Representation Learning

Multi-modal representation learning Manzoor et al. (2023) enables information fusion and transformation across different modalities (e.g., images, LiDAR point clouds, text, speech) by learning a shared representation space. In autonomous driving, approaches like Zhang et al. (2025); Liu et al. (2023); Lu et al. (2024) employ network designs such as sparse transformers and feature pyramids to learn fused multi-modal representations from LiDAR point clouds and camera images, significantly enhancing vehicles' environmental perception capabilities. Knowledge distillation serves as a common method for cross-modal knowledge transfer, approaches like Zhou et al. (2023); Wang et al. (2024); Chen et al. (2022) apply various distillation losses, including dense distillation loss, relative relation distillation loss, and response distillation loss, between multi-modal features to achieve mutual enhancement of multi-modal information, thereby improving task performance. This paper generates the common representation from the local representations of each modality using a feature pyramid network, while introduces a multi-dimensional alignment loss composed of distribution alignment loss, structural alignment loss, and pragmatic alignment loss during training to enable more effective alignment of local representations to the multi-modal common representation.

## 3 Method

### 3.1 Framework

NegoCollab achieves heterogeneous collaboration through the negotiated common representation. As is shown in Figure 2, by introducing plug-and-play sender-receiver pairs for each agent, the mutual conversion of features between the local representation space and the common representation space is achieved, thereby eliminating domain gap. Let $\mathcal{H}_*^{(m)}(\cdot)$ denote the model used by the agent with modality $m$, where * denotes the name of any module in the model, $m \in \{1, 2, ..., M\}$ and $M$ is the total number of modalities (specific sensor and perception encoder constitute a modality). The structures of the sender and receiver, as well as the collaboration process, are described below:

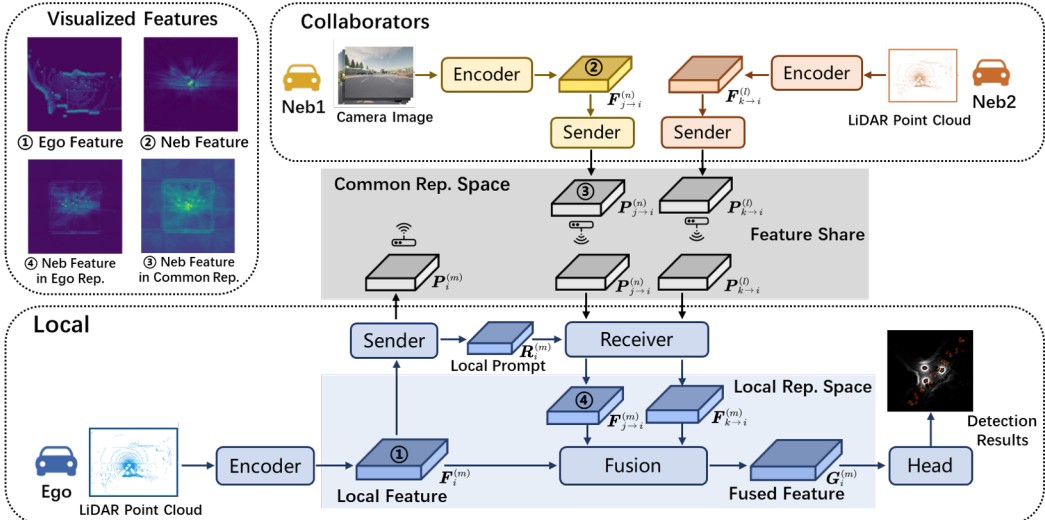

Figure 2: Overview of NegoCollab. Each agent shares features in the negotiated common representation space. Through the sender-receiver pairs, the features are mutually converted between local representation space and the common representation space, thereby enabling the mutual transformation of features across modalities and eliminating domain gaps.

#### 3.1.1 Sender

The sender's role is to transform features from the local representation space to the common representation space, consisting of two modules: recombiner and aligner. The recombiner employs a ConvNeXtLiu et al. (2022) structure to enhance local features beneficial for collaboration. It also includes a size-channel alignment module to adjust the dimensions and channels of local features to standard settings. The aligner uses a fused axial attention Xu et al. (2022) to capture both global and local dependencies within features, thereby mapping features from the local representation space to the common representation space.

During collaboration, for agent $i$ with modality $m$ in the scene, where $N$ is the total number of agents, its local observation $O_i$ is first encodes by a perception encoder $\mathcal{H}_{\text{encoder}}^{(m)}$ to extract initial feature $F_i^{(m)} = \mathcal{H}_{\text{encoder}}^{(m)}(O_i)$. Then the initial feature are transformed into the common representation space by the sender and shared with the collaborators, formalized process is as follows:

$$R_i^{(m)} = \mathcal{S}_{\text{recombiner}}^{(m)}\left(F_i^{(m)}\right) \tag{1}$$

$$P_i^{(m)} = \mathcal{S}_{\text{aligner}}^{(m)}\left(R_i^{(m)}\right) \tag{2}$$

#### 3.1.2 Receiver

The role of receiver is to transform the received features from collaborators from the common representation space back to the local representation space, consisting of two modules: converter and recombiner. The converter adopts a fused axial attention to transform features from the common representation space to the local representation space. The query vector $Q$ in its input comes from the output $R_i^{(m)}$ of the $\mathcal{S}_{\text{recombiner}}^{(m)}(\cdot)$ in sender, providing local modality guidance information for the transformation of collaborative features. The recombiner employs a ConvNeXt architecture to further reorganize and adjust local feature information, enabling adaptation to the local fusion module.

Let $P_{j\to i}^{(n)}$ denote the features received from collaborator $j \in \mathcal{N}_i$ with modality $n$, where $\mathcal{N}_i$ represents the set of collaborators for agent $i$. The formalized process of the receiver is as follows:

$$T_{j\to i}^{(m)} = \mathcal{R}_{\text{converter}}^{(m)}\left(R_i^{(m)}, P_{j\to i}^{(n)}\right), \tag{3}$$

$$F_{j\to i}^{(m)} = \mathcal{R}_{\text{recombiner}}^{(m)}\left(T_{j\to i}^{(m)}\right). \tag{4}$$

Finally, the transformed features $F_{j \to i}^{(m)}$ from the collaborator and the local initial feature $F_i^{(m)}$ are fused to obtain the fused feature $G_i^{(m)}$. The fused feature is then processed by the task head to obtain the task result $D_i^{(m)}$, completing the process of collaborative perception. Formalized process is as follows:

$$G_i^{(m)} = \mathcal{H}_{\text{fuse}}^{(m)} \left( F_i^{(m)}, F_{j \to i}^{(m)} \right), \tag{5}$$

$$D_i^{(m)} = \mathcal{H}_{\text{head}}^{(m)} \left( G_i^{(m)} \right). \tag{6}$$

## 3.2 Training

In the heterogeneous collaboration method based on common representations, whether the domain converter can effectively achieve the mutual conversion of features between local representation space and the common representation space is of crucial importance to the collaboration performance. To address this, we introduce a negotiator that generates the common representation from each modality's local representations, thereby reducing the inherent domain gap between the common representation and local representations and consequently decreasing the training difficulty for sender-receiver pairs. The training process consists of two stages: The objective of the first stage is to negotiate common representations and to enable the sender-receiver to transform features from the local representation to and from the common representation. The training loss includes two components: cyclic distribution consistency loss and multi-dimensional alignment loss. The objective of the second stage is to adapt the framework to downstream collaborative tasks. This is achieved by fine-tuning the receiver parameters using the collaborative task loss. Detailed training procedure is described below, diagram is provided in the appendix.

### 3.2.1 Pairwise Local Representation Extraction

Since both the distribution cycle-consistent loss and multi-dimensional alignment loss require paired representations for computation, we provide each modality's observation encoder with observation data from the same perspective during training. Let $O = \{O_1, O_2, ..., O_N\}$ denote the observation data from all $N$ perspectives in the scene. At the start of training, we first input the observation data $O$ into each modality's perception encoder to obtain the initial local representations for each modality. Then, we use a resizer to align the sizes and channels of these representations to the standard configuration. The formalized process is as follows:

$$F^{(m)} = \mathcal{H}_{\text{encoder}}^{(m)} \left( O \right), \tag{7}$$

$$U^{(m)} = \mathcal{H}_{\text{resizer}}^{(m)} \left( F^{(m)} \right). \tag{8}$$

### 3.2.2 Generates Common Representation by Negotiator

After obtaining the standardized local representations $U^{(m)}$, we use the negotiator to generate the common representation from each modality's local representations. The main structure of the negotiator is a feature pyramid network, where each level contains an estimator to evaluate the contribution of each modality's representation to the common representation at that level, detailed illustrations is in appendix. Specifically, a pyramid network is first used to extract multi-level features $U_l^{(m)}$ from $U^{(m)}$, and the corresponding estimators at each level is used to evaluate their contribution weights to the common representation, producing an importance matrix $C_l^{(m)}$. Next, at each level, the $U_l^{(m)}$ and $C_l^{(m)}$ from all modalities are multiplied and then averaged to obtain the common representations $P_l$ for that level. Subsequently, all $P_l$ are concatenated after alignment through upsampling. Afterward, their sizes and channels are restored to standard settings via a shrink header, yielding the common representation P. Let the input $U_0^{(m)}$ at level 0 of the pyramid be $U^{(m)}$. The

formalized process of the negotiator is as follows:

$$U_l^{(m)} = \mathcal{N}_{\text{layer}_l}\left(U_{l-1}^{(m)}\right), \quad l = 1, 2, ..., L, \tag{9}$$

$$C_l^{(m)} = \mathcal{N}_{\text{estimator}_l}\left(U_l^{(m)}\right), \quad l = 1, 2, ..., L, \tag{10}$$

$$P_l = \text{sum}\left(\left\{U_l^{(m)} \odot C_l^{(m)}\right\}_{m=0}^{M}\right)/M, \tag{11}$$

$$P = \text{contact}\left([u_l\left(P_l\right)]_{l=0}^{L}\right), \tag{12}$$

$$P = \mathcal{N}_{\text{shrink\_header}}\left(P\right), \tag{13}$$

where $l$ denotes the pyramid level, $m$ represents the modality of the representation, $\odot$ indicates the Hadamard product, and $u_l\left(\cdot\right)$ stands for the upsampling operation.

Next, the common representation $P$ is fed into each modality's receiver and transformed back to the local representation $L^{(m)}$:

$$T^{(m)} = \mathcal{R}_{\text{converter}}^{(m)}\left(R^{(m)}, P\right), \tag{14}$$

$$L^{(m)} = \mathcal{R}_{\text{recombiner}}^{(m)}\left(T^{(m)}\right). \tag{15}$$

At this stage, the cyclic distribution consistency loss can be computed as follows:

$$\mathcal{L}_{cycle}^{(m)} = \left\|F^{(m)} - L^{(m)}\right\|_2^2 + \beta\left\|Std\left(F^{(m)}\right) - Std\left(L^{(m)}\right)\right\|_2^2. \tag{16}$$

Through the constraint of cyclic distribution consistency loss, the information loss during mutual transformation between the common representation and local representations is minimized, thereby effectively reducing the inherent domain gap between them.

### 3.2.3 Multi-dimensional Information Alignment

We impose a multi-dimensional alignment loss constraint between the common representation output by senders and the negotiator. This constraint consists of three components: distribution consistency loss, structural alignment loss, and pragmatic alignment loss. Its purpose is to fully distill the representational information from the multimodal common representations into the sender, thereby facilitating the transformation from local representations to the common representation. The formulation process is as follows:

First, we use the sender to transform the local representations $F^{(m)}$ into common representation:

$$R^{(m)} = \mathcal{S}_{\text{recombiner}}^{(m)}\left(F^{(m)}\right), \tag{17}$$

$$P^{(m)} = \mathcal{S}_{\text{aligner}}^{(m)}\left(R^{(m)}\right). \tag{18}$$

Next, we compute the multi-dimensional alignment loss between common representations $P^{(m)}$ output by senders and the common representation $P$ output by the negotiator. This loss enforces distribution consistency, structural consistency, and pragmatic consistency between $P^{(m)}$ and $P$. Here, distribution consistency ensures that the statistical characteristics of the representations match. This is achieved by applying a distribution alignment loss that constrains $P^{(m)}$ and $P$ to have identical means and standard deviations, computed as follows:

$$\mathcal{L}_{uni-dis}^{(m)} = \left\|P^{(m)} - P\right\|_2^2 + \alpha\left\|Std\left(P^{(m)}\right) - Std\left(P\right)\right\|_2^2. \tag{19}$$

Structural consistency ensures that the spatial relationships between scene components remain coherent across representations. This is achieved by enforcing consistent relative relationships between different parts of samples. Specifically, for each sample $s$, where $s \in \{1, 2, ..., S\}$ and $S$ is the total number of samples, we consider the interrelationships among 9 key points $\{(x_i, y_i)\}_{i=1}^{9}$. Features of keypoints are collected from samples sampled from the common representations $P^{(m)}$

and $P$, and the relative relation matrix of sample is obtained by calculate the similarity between keypoints:

$$M_{i,j}^{P_s^{(m)}} = \mathcal{C}\left(P_s^{(m)}\left(x_i, y_i\right), P_s^{(m)}\left(x_j, y_j\right)\right),\tag{20}$$

$$M_{i,j}^{P_s} = \mathcal{C}\left(P_s\left(x_i, y_i\right), P_s\left(x_j, y_j\right)\right),\tag{21}$$

where $1 \leqslant i, j \leqslant 9$, and $\mathcal{C}\left(\cdot, \cdot\right)$ denotes the cosine similarity between elements. The relative relationship matrices of all sample pairs in $P^{(m)}$ and $P$ are made consistent to achieve structural consistency. The structural alignment loss is calculated as follows:

$$\mathcal{L}_{uni-stru}^{(m)} = \sum_{s=1}^{S}\left(\sum_{1 \leqslant i,j \leqslant 9}|M_{i,j}^{P_s^{(m)}} - M_{i,j}^{P_s}|\right)/81.\tag{22}$$

Pragmatic consistency refers to the consistent organization of foreground information in the representation space. It is achieved by training a shared 2D occupancy prediction network for the common representations $P^{(m)}$ and $P$, which aligns the organization of foreground information through reverse alignment. Let $\mathcal{N}(\cdot)$ denote the shared 2D occupancy prediction network, and $Y$ be the 2D occupancy labels corresponding to observation data $O$. The pragmatic alignment losses for $P^{(m)}$ and $P$ are computed as follows, respectively:

$$\mathcal{L}_{uni-pragma}^{(m)} = L_{focal}\left(\mathcal{N}\left(P^{(m)}\right), Y\right),\tag{23}$$

$$\mathcal{L}_{pragma}^{(p)} = L_{focal}\left(\mathcal{N}(P), Y\right),\tag{24}$$

where $L_{focal}$ is the focal loss Lin et al. (2017).

Then, the multi-dimensional alignment loss of modality $m$ is obtained by summing the distribution consistency loss, the structural consistency loss, and the pragmatic consistency loss:

$$\mathcal{L}_{uni}^{(m)} = \lambda_d \mathcal{L}_{uni-dis}^{(m)} + \lambda_s \mathcal{L}_{uni-stru}^{(m)} + \lambda_p \mathcal{L}_{uni-pragma}^{(m)}.\tag{25}$$

Finally, the first-stage training loss is calculated as a weighted sum of the distribution cycle-consistent losses, the multi-dimensional alignment losses from all modalities, and the pragmatic alignment loss of the common representation $P$:

$$\mathcal{L}_{stage1} = \lambda_a \mathcal{L}_{pragma}^{(p)} + \sum_{m=1}^{M}\lambda_c \mathcal{L}_{cycle}^{(m)} + \lambda_u \mathcal{L}_{uni}^{(m)}.\tag{26}$$

### 3.2.4 Task Adaption

To enable the receiver to focus on restoring information beneficial for collaboration, we fine-tune the receivers of each modality using the downstream collaborative task loss for the second stage of training. During this process, the data loading method and feature flow are identical to those during inference (Section 3.1), the parameters of the senders are fixed, and the loss is calculated as follows:

$$\mathcal{L}_{stage2} = \sum_{i=1}^{N}\mathcal{L}_{collab}\left(D_i^{(m)}, Y_i\right).\tag{27}$$

Here, $\mathcal{L}_{collab}$ is the collaborative task loss, $D_i^{(m)}$ is derived from Equation 6 and represents the task prediction output by the collaborative model, while $Y_i$ denotes the task label for agent $i$.

## 4 Experiment

### 4.1 Settings

We configure four collaborating agents m1, m2, m3, m4 and one protocol agent in the scenario. Among them, the protocol agent, m1, and m3 are equipped with LiDAR sensors, while m2 and m4

are equipped with cameras. The perception encoders used by m1 and m3, as well as those used by m2 and m4, are different. Detailed configurations are provided in the Appendix.

To evaluate the performance of the common representation and its generalization capability to new agents, we form an initial collaborative alliance between agent m1 and agent m2, from which the common representation are negotiated. Agents m3 and m4 are newly added agents that align their features with the common representation. The training process consists of three stages:

**Step 0:** *Homogeneous collaborative training.* For each of the 4 agent types, train a homogeneous collaborative perception model.

**Step 1:** *Initial alliance negotiation.* Following the method in Section 3.2, the training is conducted in two stages. In the first stage, sender-receiver pairs are introduced to m1 and m2, respectively. A common representation is obtained through training assisted by the negotiator to complete the training of sender-receiver pairs. In the second stage, the parameters of the receivers for m1 and m2 are adjusted to adapt to the downstream collaborative task. During the training process, the parameters of the perception encoder, fusion module, and task head in the homogeneous collaborative perception model for m1 and m2 are frozen.

**Step 2:** *New agent joins.* The training when new agents m3 and m4 join is also divided into two stages. The loss calculation in the first stage is the same as in Section 3.2, but the common representation is obtained directly from the perception encoders of m1 and m2 and the negotiator. The collaborative task loss in the second stage is calculated as the collaborative task loss of the new agents and the existing agents in the alliance. During the training process, the parameters of the negotiator, the perception encoders of m1 and m2, and the parameters of the homogeneous collaborative model for m3 and m4 are frozen. Specific illustration is provided in the appendix.

Table 1: Performance comparison of heterogeneous collaboration on OPV2V-H. "NegoCollab-P", "MPDA-P" and "PnPDA-P" after added "-P" are special implementations of the corresponding methods, which feature sharing is achieved by using the representation of the protocol agent as the common representation.

| Metric | | AP@0.5 | | | | AP@0.7 | | | |
|---|---|---|---|---|---|---|---|---|---|
| Agent Types | | m1m2 | m1m3 | m2m4 | All | m1m2 | m1m3 | m2m4 | All |
| No Fusion | | 0.482 | 0.794 | 0.221 | 0.480 | 0.350 | 0.687 | 0.106 | 0.342 |
| One-to-one Adaptation | MPDA | 0.815 | 0.922 | 0.520 | 0.512 | 0.692 | 0.850 | 0.331 | 0.435 |
| | PnPDA | 0.865 | 0.949 | 0.532 | 0.494 | 0.755 | 0.903 | 0.351 | 0.424 |
| Align to Common | MPDA-P | 0.561 | 0.811 | 0.354 | 0.465 | 0.409 | 0.697 | 0.173 | 0.353 |
| | PnPDA-P | 0.552 | 0.875 | 0.365 | 0.434 | 0.447 | 0.805 | 0.216 | 0.346 |
| | STAMP | 0.545 | 0.770 | 0.264 | 0.382 | 0.448 | 0.708 | 0.134 | 0.286 |
| | NegoCollab-P | 0.792 | 0.772 | 0.499 | 0.676 | 0.615 | 0.710 | 0.289 | 0.457 |
| | NegoCollab | **0.872** | **0.911** | **0.512** | **0.745** | **0.765** | **0.854** | **0.319** | **0.555** |

## 4.2 Quantitative Analysis

**Performance of heterogeneous collaboration.** We evaluated each method on the OPV2V-H Lu et al. (2024), V2V4Real Xu et al. (2023c), and DAIR-V2X Yu et al. (2022) datasets, as shown in Table 1 and Table 2. Since the common representation of MPDA-P, PnPDA-P, and STAMP are all derived from the single-modality protocol agent, for fair comparison, we implement NegoCollab-P, which derives the common representation from the protocol agent. In Table 1, the columns m1m2, m1m3, m2m4, and m1m2m3m4 correspond to the performance of: initial alliance agents, heterogeneous LiDAR agents, heterogeneous camera agents, and all agent types collaborative, respectively. The results demonstrate that among heterogeneous collaboration methods based on common representation, NegoCollab achieves the best performance in all test conditions. Compared with one-to-one adaptation methods, NegoCollab also maintains optimal collaborative performance when agents m1 and m2 within the initial alliance collaborated. For collaboration with new agents m3 and m4, although m3 and

Table 2: Performance comparison of heterogeneous collaboration on real-world datasets V2V4Real and DAIR-V2X, with collaborating agents being m1 and m3, m1 and m2 respectively.

| Methods | | V2V4Real | | DAIR-V2X | |
|---|---|---|---|---|---|
| | | AP@0.5 | AP@0.7 | AP@0.5 | AP@0.7 |
| No Fusion | | 0.504 | 0.358 | 0.329 | 0.219 |
| One to one Adaption | MPDA | 0.613 | 0.400 | 0.344 | 0.235 |
| | PnPDA | 0.598 | 0.385 | 0.443 | 0.277 |
| Align to Common | MPDA-P | 0.467 | 0.334 | 0.258 | 0.211 |
| | PnPDA-P | 0.485 | 0.324 | 0.230 | 0.192 |
| | STAMP | 0.466 | 0.345 | 0.299 | 0.161 |
| | NegoCollab-P | 0.482 | 0.333 | 0.376 | 0.195 |
| | NegoCollab | **0.605** | **0.397** | **0.397** | **0.241** |

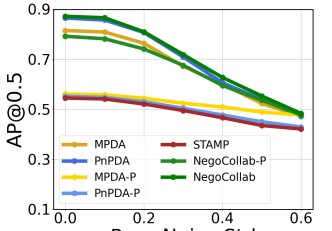 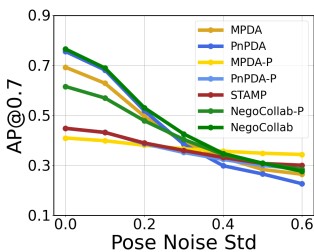 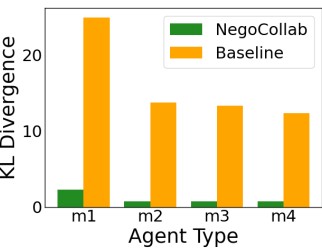

Figure 3: Robustness Analysis of Localization Errors. Pose noise is set to $\mathcal{N}\left(0, \sigma^2\right)$ on both x,y location and yaw angle. The collaborating agents are m1 and m2.

Figure 4: Comparison of domain gaps between local and common representation.

m4 did not participate in the negotiation process of the common representation, their collaborative performance is slightly lower than that of one-to-one adaptation methods, but still achieves competitive results. This strongly demonstrates NegoCollab's superior performance and the excellent adaptability of the common representation to new agents. Additionally, the results in Table 2 show that NegoCollab also has excellent heterogeneous collaboration performance in real-world environments.

Table 3: Comparison of homogeneous collaboration performance when sharing features in the common representation space. "Local" denotes direct feature sharing through local representation spaces. Evaluation was conducted on the OPV2V-H dataset.

| Metric | AP@0.5 | | | | AP@0.7 | | | |
|---|---|---|---|---|---|---|---|---|
| Agent Type | m1 | m2 | m3 | m4 | m1 | m2 | m3 | m4 |
| Local | 0.952 | 0.540 | 0.930 | 0.497 | 0.919 | 0.378 | 0.886 | 0.322 |
| MPDA-P | 0.837 | 0.515 | 0.804 | 0.439 | 0.712 | 0.305 | 0.684 | 0.230 |
| PnPDA-P | 0.950 | 0.545 | 0.926 | 0.499 | 0.910 | 0.362 | 0.883 | 0.309 |
| STAMP | 0.945 | 0.555 | 0.925 | 0.497 | 0.892 | 0.373 | 0.868 | 0.304 |
| NegoCollab-P | 0.951 | 0.566 | 0.932 | 0.513 | 0.916 | 0.378 | 0.881 | 0.317 |
| NegoCollab | **0.953** | **0.570** | **0.933** | **0.521** | **0.911** | **0.385** | **0.888** | **0.317** |

**Performance of homogeneous collaboration.** Table 3 presents the homogeneous collaboration performance of different methods when using the common representation to share feature. As shown, NegoCollab achieves the best performance among all methods. For agents m1, m3, and m4, it even surpasses the original homogeneous collaboration performance. This improvement stems from the multi-dimensional alignment loss distilling multi-modal knowledge from common representation into local senders, thereby enhancing the feature's representational capacity.

**Comparison of domain gaps.** To validate the effectiveness of the negotiator in reducing domain gaps, we employ KL divergence Kullback and Leibler (1951) to measure the domain gap between common

representation and local representations of each modality across different methods. Comparision are illustrated in Figure 4. Since MPDA-P, PnPDA-P, and STAMP all use the representation of the protocol agent as the common representation, they are aggregated as the 'Baseline' in the figure. It can be seen that the domain gap between the common representation generated by the negotiator and each local representation is significantly reduced. Compared to the method of directly designating the representation of the protocol agent as the common representation, the domain gap measured by KL divergence is reduced by an average of approximately 93.5

**Localization error robustness.** We introduced Gaussian noise to the accurate poses to evaluate the noise robustness of each method, as shown in Figure 3. The results demonstrate that under various error conditions, NegoCollab maintained superior performance on the AP@0.5 evaluation metric.

## 4.3 Ablation Study

**Negotiating from different initial alliances.** In practical applications, heterogeneous agents form multiple collaborative groups based on collaboration needs Gao et al. (2025), using different common representations for information sharing within each group. NegoCollab's negotiation-based mechanism enables the free selection of agents from a collaborative group to negotiate the common representation, thereby providing more diverse and reliable common representation. To further explore how to negotiate a better common representation, we investigate the impact of common representations negotiated from different initial alliances on collaborative performance. Two key observations are summarized. with detailed content and experimental results provided in the Appendix.

**Training Setting Ablation.** We conducted ablation studies on the negotiator and the multi-dimensional alignment loss within the training setup on the OPV2V-H dataset. The results before adaption for the downstream collaborative task are presented in Table 4. Under the initial setup, the multi-dimensional alignment loss includes only the distribution alignment loss, without assistance from the negotiator during training. The common representation is obtained by directly constraining the outputs of each modality's senders to be consistent through the alignment loss. A comparison between the upper and lower sections of the table demonstrates that negotiate common representation by the negotiator effectively enhanced the performance in heterogeneous collaboration. The performance

Table 4: Ablation study of the traning setting. The collaborating agents are m1 and m2.

| Nego | uni-stru | uni-pragma | AP@0.5 | AP@0.7 |
|---|---|---|---|---|
| | | | 0.617 | 0.490 |
| | ✓ | | 0.609 | 0.485 |
| | | ✓ | 0.627 | 0.499 |
| | ✓ | ✓ | **0.635** | **0.508** |
| ✓ | | | 0.609 | 0.496 |
| ✓ | ✓ | | 0.655 | 0.532 |
| ✓ | | ✓ | 0.671 | 0.538 |
| ✓ | ✓ | ✓ | **0.711** | **0.566** |

improvements observed in the "uni-stru" and "uni-pragma" columns indicate that the structural and pragmatic alignment losses effectively facilitated the transformation of local representations into the common representation.

## 5 Conclusion

This paper proposes NegoCollab, a heterogeneous collaboration method based on negotiating common representation. NegoCollab uses a negotiator to generate the common representation from the local representations of each modality's agent, effectively reducing the domain gap between the common representation and the local representations. Furthermore, by introducing a multi-dimensional alignment loss, it effectively promotes better alignment of the local representations to the multi-modal common representation. Evaluation results from both simulated and real-world environments collectively demonstrate the outstanding heterogeneous collaboration performance of NegoCollab. A limitation of NegoCollab is that once the common representation is negotiated, it becomes fixed. Aligning new agents to this pre-negotiated common representation inevitably leads to greater information loss. We will explore methods to make the common representation generalize better to new agents in the future.

# 6  Acknowledgement

This work was supported in part by the National Key Research and Development Program of China under Grant 2023YFB4301900, in part by the Natural Science Foundation of China under Grant 62272053 and Grant 62472048, in part by the Beijing Nova Program under Grant 20230484364, and in part by Beijing Natural Science Foundation under Grant L242081.

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

# A Detailed Setup of Experiment

## A.1 Dataset

**OPV2V-H.** OPV2V-H Lu et al. (2024) dataset contains 73 scenes covering 6 road types across 9 cities. Each Connected Autonomous Vehicle(CAV) in the scenes is equipped with one 16-channel, one 32-channel, and one 64-channel LiDAR, along with 4 monocular cameras and 4 depth cameras. The dataset comprises 36K frames of LiDAR point clouds, 12K frames of RGB camera images, 12K frames of depth camera images, and 230K annotated 3D bounding boxes.

**DAIR-V2X.** DAIR-V2X Yu et al. (2022) is a real-world collaborative perception dataset. The dataset has 9K frames featuring one vehicle and one roadside unit (RSU), both equipped with a LiDAR and a 1920x1080 camera. RSU' LiDAR is 300-channel while the vehicle's is 40-channel.

**V2V4Real.** V2V4Real Xu et al. (2023c) is a real-world Vehicle-to-Vehicle (V2V) cooperative perception dataset. The dataset includes 20,000 LiDAR scans and 240,000 annotated 3D bounding boxes across five vehicle classes. It supports benchmarks for three key task: 3D object detection, object tracking, and Sim2Real domain adaptation-enabling evaluation with state-of-the-art models.

## A.2 Training Setup

We conducted testing and training using a single RTX 4090 GPU, with an initial learning rate of 0.001 and Adam optimizer for parameter adjustment. The first training phase required approximately 4-12 GPU hours with about 23GB memory usage, while the second phase took around 2-5 GPU hours consuming approximately 14GB memory. The exact values depend on the specific agent model architecture.

## A.3 Detailed Configuration of Agents

Section 4.1 mentions 4 types of agents m1, m2, m3, and m4, as well as protocol agents. The detailed configurations of their sensors and perception encoders are shown in Table 5.

Table 5: Settings for sensors and perception encoders of agents.

| Agent Type | Sensor | Perception Encoder |
|---|---|---|
| **Protocol** | LiDAR of 64-channel | PointPillars |
| **m1** | LiDAR of 64-channel | PointPillars |
| **m2** | Camera, resize img. to height 384 px | Lift-Splat w. EfficientNet as img. encoder |
| **m3** | LiDAR of 32-channel | SECOND |
| **m4** | Camera, resize img. to height 336 px | Lift-Splat w. ResNet50 as img. encoder |

# B More Experiments

## B.1 Negotiating from Different Initial Alliances

We investigate the impact of negotiating common representation from different initial alliances on collaborative performance, as shown in Table 6. It can be observed that in the heterogeneous collaboration scenario, for common representations negotiated from different initial alliances, when the participating agents are consistent with those in their initial alliance, the optimal performance is achieved in the corresponding collaboration scenario. In homogeneous collaboration, compared to directly sharing features using local representations, sharing features using different common representations results in nearly unchanged collaboration performance for agents m1 and m3, and even better performance for agents m2 and m4. This is because the multi-dimensional alignment loss effectively distills multimodal knowledge from the common representation into the local senders and receivers, thereby enhancing the performance of the representations.

Furthermore, we derive two key observations from the results in Table 6b:

Table 6: Performance comparison when negotiating common representations from different initial alliances. The "Initial Alliance" column indicates the agents in the initial alliance, while the remaining agents are new agents. The training process is the same as that described in Section 4.1.

(a) Performance of heterogeneous collaboration

| Initial Alliance | AP@0.5 | | | | | AP@0.7 | | | | |
|---|---|---|---|---|---|---|---|---|---|---|
| | m1m2 | m3m4 | m1m3 | m2m4 | All | m1m2 | m3m4 | m1m3 | m2m4 | All |
| Protocol | **0.792** | 0.785 | 0.772 | **0.499** | 0.676 | **0.615** | 0.564 | 0.710 | **0.289** | 0.457 |
| m1m3 | 0.869 | 0.832 | **0.951** | 0.484 | **0.830** | 0.761 | 0.720 | **0.904** | 0.280 | **0.718** |
| m1m2 | **0.872** | 0.770 | 0.911 | **0.512** | 0.745 | **0.759** | 0.578 | 0.805 | **0.319** | 0.555 |
| m3m4 | 0.727 | 0.840 | **0.914** | 0.506 | **0.737** | 0.550 | 0.726 | **0.840** | 0.289 | **0.562** |

(b) Performance of homogeneous collaboration

| Initial Alliance | AP@0.5 | | | | AP@0.7 | | | |
|---|---|---|---|---|---|---|---|---|
| | m1 | m2 | m3 | m4 | m1 | m2 | m3 | m4 |
| Local | 0.952 | 0.540 | 0.930 | 0.497 | 0.919 | 0.378 | 0.886 | 0.322 |
| Protocol | 0.951 | 0.566 | 0.932 | 0.513 | 0.916 | 0.378 | 0.881 | 0.317 |
| m1m3 | 0.953 | 0.568 | 0.932 | 0.512 | 0.913 | 0.378 | 0.882 | 0.315 |
| m1m2 | 0.953 | 0.570 | 0.933 | 0.521 | 0.911 | 0.385 | 0.888 | 0.317 |
| m3m4 | 0.953 | 0.575 | 0.932 | 0.511 | 0.914 | 0.391 | 0.883 | 0.313 |

- Common representations negotiated from more types of agents demonstrate superior performance. As shown in rows 1 ("Protocol") and 3 ("m1m2") of Table 6b, compared to representation negotiated solely from LiDAR-equipped protocol agent, those obtained from the initial alliance comprising both LiDAR-equipped agent m1 and camera-equipped agent m2 achieve better performance in m1m2, m1m3, m2m4, and all types of agent collaboration scenarios.

- Common representations negotiated from agents with superior perception encoder performance yield better results. As evidenced by rows 4 ("m3m4") and 2 ("m1m3") in Table 6b, representations negotiated from agents m1 and m2 - which have better perception performance when using identical sensors - demonstrate stronger generalization to new agents m3 and m4. Conversely, representations derived from agents m3 and m4 with inferior perception exhibit degraded performance when collaborating with new agents m1 and m2. Therefore, when sensors are identical, agents with better-performing perception encoders should be prioritized to form the initial alliance.

## B.2 Comparison with Late Fusion

We further contrast the performance of NegoCollab with late fusion, as shown in Table **?**. Late fusion generally performs better when there is no localization error in different scenario. This is because, compared to intermediate fusion, late fusion directly merges detection results, which can mitigate the impact of model heterogeneity on collaboration. As the localization error increases, the performance of late fusion declines significantly. In contrast, NegoCollab-P and NegoCollab, based on the intermediate fusion, demonstrate greater robustness and achieve performance substantially superior to late fusion. This is because feature-level fusion combines the features from collaborative agents based on semantic similarity, which can mitigate the impact of locaization error to some extent. Since localization errors are almost unavoidable in practical scenarios, the more robust NegoCollab exhibits stronger practicality.

Table 7: Performance Comparison with Late Fusion under different localization error conditions. The agent positions are perturbed with Gaussian noise, where $\sigma$ represents the standard deviation of the Gaussian noise. The "Avg. Inc." column corresponds to the increase in the average evaluation results of NegoCollab and NegoCollab-P across various collaborative scenarios under different noise conditions, compared to late fusion.

| $\sigma$ | Agent Types | AP@0.5 | | | | Avg. Inc. | AP@0.7 | | | | Avg. Inc. |
|---|---|---|---|---|---|---|---|---|---|---|---|
| | | m1m2 | m1m3 | m2m4 | m1m2 m3m4 | | m1m2 | m1m3 | m2m4 | m1m2 m3m4 | |
| 0.0 | Late Fusion | 0.873 | 0.952 | 0.482 | 0.854 | - | 0.743 | 0.893 | 0.290 | 0.725 | - |
| | NegoCollab-P | 0.792 | 0.772 | 0.499 | 0.676 | -13.3% | 0.615 | 0.710 | 0.289 | 0.457 | -21.9% |
| | NegoCollab | 0.872 | 0.911 | 0.512 | 0.745 | -3.8% | 0.765 | 0.854 | 0.319 | 0.555 | -0.06% |
| 0.3 | Late Fusion | 0.564 | 0.626 | 0.299 | 0.543 | - | 0.201 | 0.271 | 0.077 | 0.197 | - |
| | NegoCollab-P | 0.676 | 0.711 | 0.391 | 0.591 | +16.6% | 0.403 | 0.527 | 0.149 | 0.388 | +96.6% |
| | NegoCollab | 0.719 | 0.837 | 0.387 | 0.616 | +25.9% | 0.425 | 0.582 | 0.146 | 0.365 | +103.4% |
| 0.6 | Late Fusion | 0.278 | 0.328 | 0.154 | 0.264 | - | 0.115 | 0.169 | 0.043 | 0.106 | - |
| | NegoCollab-P | 0.477 | 0.574 | 0.256 | 0.500 | +79.5% | 0.283 | 0.397 | 0.099 | 0.353 | +161.1% |
| | NegoCollab | 0.483 | 0.693 | 0.229 | 0.462 | +82.3% | 0.276 | 0.427 | 0.086 | 0.292 | +149.7% |

## B.3 Component Ablation

We conducted ablation experiments on the recombiner and aligner in the sender, the negotiator, and the local prompt on OPV2V-H, as shown in Table 8. It can be seen that NegoCollab achieves optimal performance when the recombiner and aligner are set to Convext and FAX(fused axial attention), respectively. This is because we divide the feature transformation process into two steps: adjusts local detail information, and transforms global representation style. The characteristics of Convext and FAX are respectively more suitable for local information adjustment and representation style transformation. For the Negotiator, the FPN structure adopted in this paper achieves the best performance with the smallest parameter count, indicating that the FPN structure can better extract common representation from each modality's local representation. After using Local Prompt to guide the transformation from the common representation to local representation, the performance is significantly improved. The above results fully demonstrate the rationality of the component design in NegoCollab.

## C Additional Illustrations

### C.1 Training Process of Initial Alliance Negotiation

Figure 5 illustrates the first-stage training process when the initial alliance negotiates the common representation as described in Section 3.2. The specific steps are as follows:

- The perception encoder of each modalitiy's agent is fed with observational data from the same perspective, encoding them into paired initial local representations $F^{(m)}$,

- The local representations $F^{(m)}$ from each modality's agent are input into the negotiator for fusion, producing a common representation $P$,

- The common representation $P$ is fed into the receiver of each modality's agent to obtain the restored local representation $L^{(m)}$,

- The initial local representation $F^{(m)}$ of each modality's agent is input into its respective sender to yield a common representation $P^{(m)}$,

- The training loss is calculated, which includes the cyclic distribution consistency loss $\mathcal{L}_{cycle}\left(F^{(m)}, L^{(m)}\right)$ between the receiver's output, and the initial local representation $F^{(m)}$ the multi-dimensional alignment loss $\mathcal{L}_{uni}\left(P, P^{(m)}\right)$ between the common representation output by the senders and the negotiator,

Table 8: Component ablation study. The collaborating agents are m1 and m2, and the results are the performance without downstream collaborative task adaptation. The component name in bold in the settings column indicates the default configuration. Column corresponding to #Params# shows the number of parameters when the module uses the corresponding configuration. 'M' standing for 'MB'. "ResMlp" is a network with a multi-layer perceptron as its backbone. FANetYoung et al. (2022) featuring an encoder-decoder structure, which can be used to adjust the feature space.

| Components | Settings | AP@0.5 | AP@0.7 | #Params# |
|---|---|---|---|---|
| **Recombiner** | ResMlp | 0.633 | 0.510 | 0.1 M |
| | FANet | 0.649 | 0.492 | 1.7 M |
| | **Convext** | 0.711 | 0.566 | 0.3 M |
| | FAX | 0.596 | 0.487 | 0.2 M |
| **Aligner** | ResMlp | 0.697 | 0.527 | 0.1 M |
| | FANet | 0.696 | 0.563 | 1.7 M |
| | Convext | 0.702 | 0.542 | 0.3 M |
| | **Fused Axial Attention** | 0.711 | 0.566 | 0.2 M |
| **Negotiator** | ResMlp | 0.705 | 0.565 | 1.8 M |
| | Convext | 0.706 | 0.566 | 2.7 M |
| | Sparse Transformer | 0.706 | 0.564 | 2.1 M |
| | **FPN** | 0.711 | 0.566 | 1.2 M |
| **Local Prompt** | w/o | 0.672 | 0.547 | - |
| | **w** | 0.711 | 0.566 | - |

- The parameters of the negotiator, as well as the sender and receiver of each modality's agent, are iteratively updated via backpropagation.

The objective of the second-stage training is to adapt the receiver for the downstream collaborative task. During this training process, the parameters of the negotiator, the perception encoders and senders of each modality's agent are frozen. The feature flow is consistent with that during inference. The loss is computed as the collaborative loss of the agents within the initial alliance.

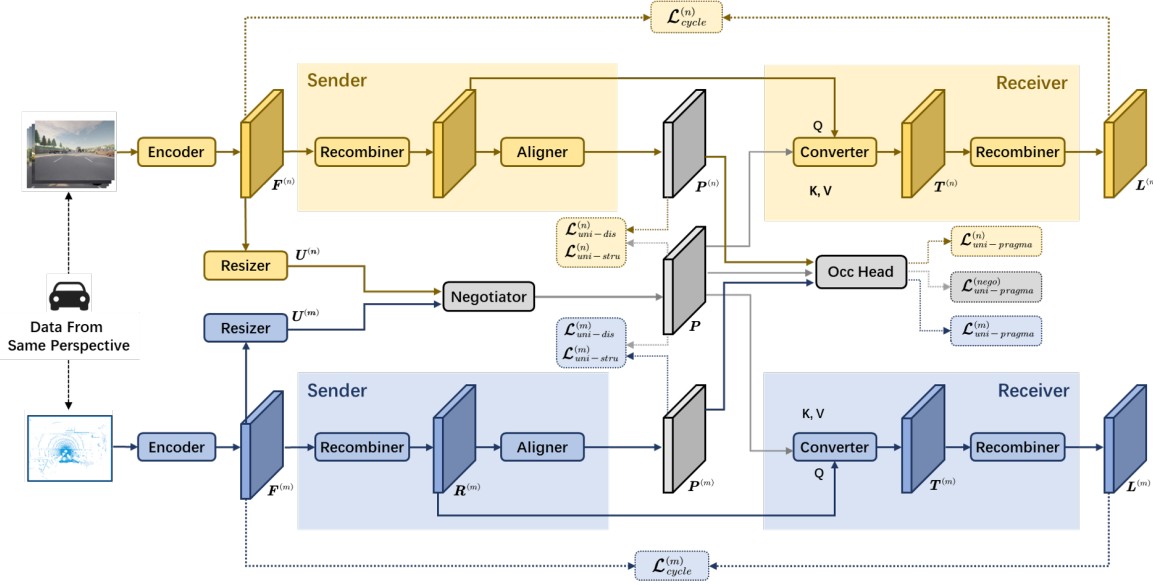

Figure 5: Training process of initial alliance negotiation.

## C.2 Training Process of New Agent Join

Figure 6 illustrates the training process of the first stage when a new agent joins. This stage aims to enable the new agent's sender and receiver to map local representations to and from the negotiated

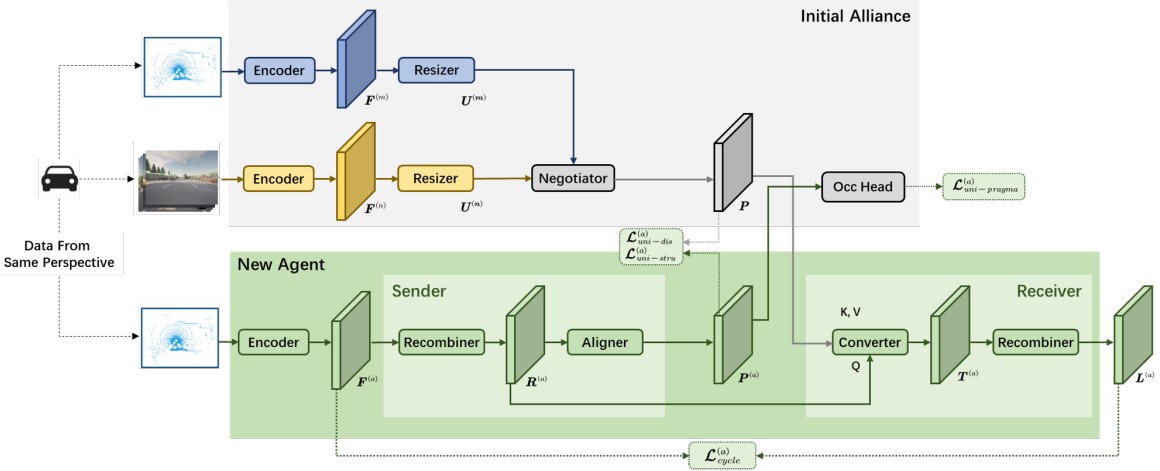

Figure 6: Training process of new agent join.

common representation, respectively. The loss calculation for this process is identical to that used during the common representation negotiation. The key difference is that the common representation is generated by leveraging the negotiator and the perception encoder of the agents within the initial alliance. The specific steps are as follows:

- Observational data from the same perspective is fed into the agents within the initial alliance and the new agent, encoding them into paired local representations $F^{(m)}$, $F^{(a)}$,

- The local representations of the agents in the initial alliance $F^{(m)}$ are input into the negotiator to produce the common representation $P$,

- The common representation $P$ is fed into the new agent's receiver to obtain the reconstructed local representation $L^{(a)}$,

- The new agent's local representation $F^{(a)}$ is input into its sender to yield a common representation $P^{(a)}$,

- The training loss is calculated, which includes the multi-dimensional alignment loss $\mathcal{L}_{uni}\left(P, P^{(a)}\right)$ between the common representation output by the negotiator and the sender of the new agent, and the cyclic distribution consistency loss $\mathcal{L}_{cycle}\left(F^{(a)}, L^{(a)}\right)$ between the receiver's output and the initial local representation,

- The parameters of the new agent's sender and receiver are iteratively updated via backpropagation, while the parameters of the negotiator and the encoders of the agents within the initial alliance remain frozen during this process.

In the second training stage, only the parameters of the new agent's receiver are adjusted, while the parameters of all other modules remain frozen. The feature flow during training is consistent with that during inference. The loss is calculated as the collaborative detection loss of the new agent and the agents within the alliance.

## C.3 Sender and Receiver

The detailed structure of the sender and receiver is shown in Figure 7. Both the sender and receiver adopt a hybrid architecture combining Transformer and ConvNeXt. The sender consists of a recombiner and an aligner, responsible for transforming local features into the common representation space. The receiver comprises a recombiner and a converter, which converts collaborators' features into the local representation space. The query vector Q in the converter is derived from the output of the recombiner in the sender.

## C.4 Negotiator

Figure 8 illustrates the process of negotiating common representation $P$ from initial local representations $F^{(m)}$ and $F^{(n)}$ of agents with modalities $m$ and $n$. Agents first extracts local representations

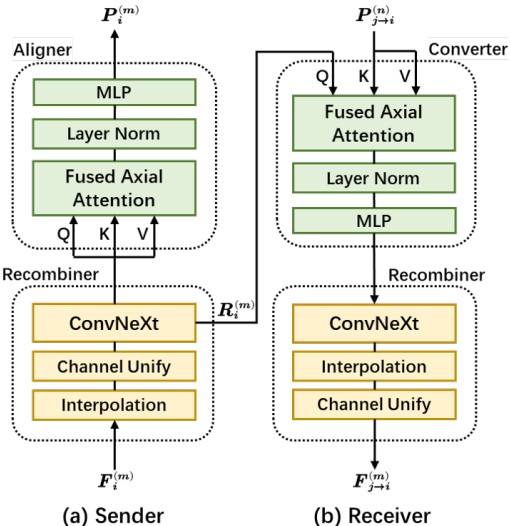

**(a) Sender**  **(b) Receiver**

Figure 7: Detailed structure of the sender and receiver. Both the sender and receiver employ a hybrid architecture integrating Transformer and ConvNeXt.

$F^{(m)}$ and $F^{(n)}$ using its native perception encoder, then aligns them to a standard size through the resizer to obtain $U^{(m)}$ and $U^{(n)}$. Subsequently, the negotiator generates the common representation $P$ from $U^{(m)}$ and $U^{(n)}$ through the following steps:

- Extract representations of each scale $U_l^{(m)}$ and $U_l^{(n)}$ from $U^{(m)}$ and $U^{(n)}$ respectively, using the pyramid network,

- At each level, employ the corresponding estimator to assess the contribution of $U_l^{(m)}$ and $U_l^{(n)}$ to the common representation, yielding the importance matrices $C_l^{(m)}$ and $C_l^{(n)}$ respectively,

- For each level, multiply $C_l^{(m)}$ with $U_l^{(m)}$, and $C_l^{(n)}$ with $U_l^{(n)}$, then average the results to obtain the level-wise common representation $P_l$,

- Upsample and concatenate all $P_l$, then restore the dimensions and channels to the standard configuration via a shrink header to produce the final common representation $P$.

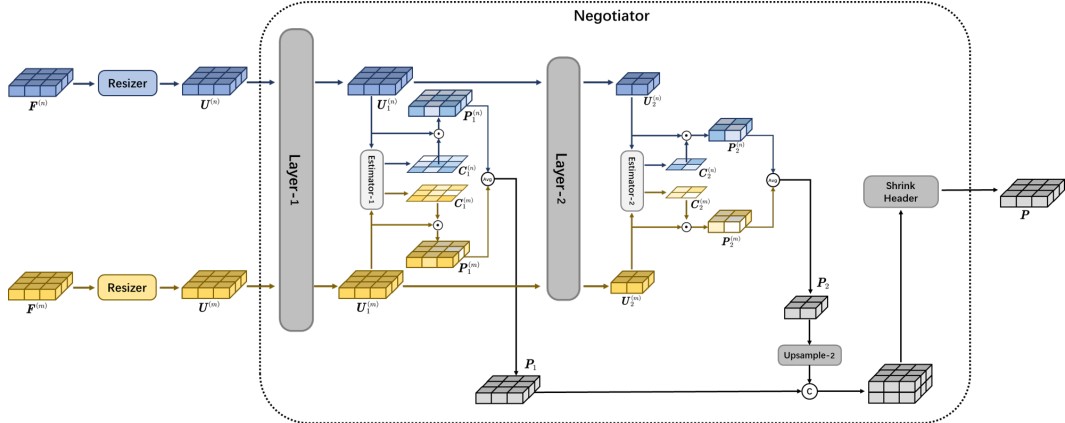

Figure 8: Architecture of negotiator. Layer-x and Estimator-x is the network of pyramid and the estimator at corresponding level.

