# OpenReview forum: "NegoCollab: A Common Representation Negotiation Approach for Heterogeneous Collaborative Perception"
_NeurIPS.cc/2025/Conference — NeurIPS 2025 poster_

### Official Review · Reviewer_fF95 · 2025-06-02

**Clarity:** 3
**Significance:** 2
**Originality:** 3
**Rating:** 4
**Confidence:** 4

**Summary:**

In this paper, the authors propose a heterogeneous collaboration approach, NegoCollab, which is based on negotiated common representation. This framework facilitates heterogeneous information exchange through a pair of sender and receiver, specifically comprising a recombiner, an aligner, and a converter. Several supervision losses are introduced to achieve bidirectional conversion between local representations and the common representation. The experimental results on the OPV2V-H dataset demonstrate the effectiveness of the proposed approach.

**Questions:**

Result-level collaboration can naturally address the heterogeneous challenge and is more computationally efficient. How about the performance comparison?

Why is the performance of NegoCollab-FT worse than one-to-one adaptation in the m1m3 and m2m4 settings? Is it due to the negotiation mechanism?

What is the coordinate system of the BEV common feature map? Does it depend on the relative positions of the collaborating vehicles?

**Ethical Concerns:**

["NO or VERY MINOR ethics concerns only"]

**Final Justification:**

The experimental results address my concerns to a large extent. I have carefully read the authors' response as well as the discussion with other reviewers. I have decided to keep my relatively positive score.

**Limitations:**

yes

**Quality:**

3

**Strengths And Weaknesses:**

The authors propose a heterogeneous collaboration approach to address the practical heterogeneity challenge in collaborative perception. The proposed NegoCollab is technically sound, training a negotiator to generate the common representation. Several supervision strategies are introduced to better align local representations with the common representation. The paper is well-organized, and the experimental analysis is comprehensive and in-depth, especially in the studies of generalization capability and negotiation mechanism.

However, there are still some concerns. The performance advantage is not significant compared with one-to-one adaptation. The proposed approach is only evaluated on a simulation dataset, whereas the heterogeneous problem in real-world scenarios is more challenging. The generalization of the common representation to new agents is still constrained by the training process.

Additionally, there are typos in lines 164 and 171. Table 1 is too wide.

---

> ### Author Rebuttal · Authors · 2025-07-28
>
> We thank the reviewer for providing valuable comments, we will address your concerns regarding the questions and show additional experimental results.
>
> >  **Q1:** Result-level collaboration can naturally address the heterogeneous challenge and is more computationally efficient. How about the performance comparison?
>
> We evaluated the performance of late fusion across different collaborative scenarios on the OPV2V-H dataset, testing its robustness against progressively increasing localization errors. Comparative results with NegoCollab and NegoCollab-P are presented in the table below:
>
> |  |            | |     AP\@0.5   |       |         | Avg. Inc. |  |    AP\@0.7   |       |         | Avg. Inc. |
> |------------|------------------|--------|-------|-------|---------|-----------|--------|-------|-------|---------|-----------|
> |    σ       | Agent Types      | m1m2   | m1m3  | m2m4  | All |           | m1m2   | m1m3  | m2m4  | All|           |
> | 0.0        | Late Fusion      | **0.873**  | **0.952** | 0.482 | **0.854**| -         | 0.743  | **0.893** | 0.290 | **0.725**| -         |
> |            | NegoCollab-P     | 0.792  | 0.772 | 0.499 | 0.676   | -13.3%    | 0.615  | 0.710  | 0.289 | 0.457   | -21.9%    |
> |            | NegoCollab    | 0.872  | 0.911 | **0.512** | 0.745 | -3.8%     | **0.765** | 0.854  | **0.319** | 0.555   | -0.06%    |
> | 0.3        | Late Fusion      | 0.564  | 0.626 | 0.299 | 0.543   | -         | 0.201  | 0.271  | 0.077 | 0.197   | -         |
> |            | NegoCollab-P     | 0.676  | 0.711 | **0.391** | 0.591 | +16.6%    | 0.403  | 0.527  | **0.149** | **0.388** | +96.6%    |
> |            | NegoCollab    | **0.719** | **0.837** | 0.387 | **0.616** | +25.9%    | **0.425** | **0.582** | 0.146 | 0.365   | +103.4%   |
> | 0.6        | Late Fusion      | 0.278  | 0.328 | 0.154 | 0.264   | -         | 0.115  | 0.169  | 0.043 | 0.106   | -         |
> |            | NegoCollab-P     | 0.477  | 0.574 | **0.256** | 0.500 | +79.5%    | 0.283  | 0.397  | **0.099** | **0.353** | +161.1%   |
> |            | NegoCollab    | **0.483** | **0.693** | 0.229 | 0.462   | +82.3%    | 0.276  | **0.427** | 0.086 | 0.292   | +149.7%   |
>
> Table 1: Performance Comparison with Late Fusion under different localization error conditions. Agent positions are perturbed with Gaussian noise of standard deviations 0.0, 0.3, and 0.6. "All" refers to the scenario when all four types of intelligent agents, m1, m2, m3, and m4, are involved in collaboration. "Avg. Inc." column in the table indicates the average performance improvement of NegoCollab and NegoCollab-P over late fusion across different collaborative scenarios under varying noise conditions.
>
>
> The results demonstrate that late fusion achieves superior performance when no localization errors exist, as it can effectively mitigate the impact of model heterogeneity. However, as localization errors progressively increase, late fusion shows significant performance degradation, while the intermediate fusion-based NegoCollab-P and NegoCollab maintain robust performance, substantially outperforming late fusion.
>
> Notably, although late fusion performs well under ideal conditions without localization errors, such perfect conditions rarely exist in real-world scenarios where localization errors are practically inevitable. Therefore, the more robust NegoCollab framework offers greater practical utility. We will include these experimental results in the revised version.
>
> &nbsp;
>
> >  **W:** The generalization of the common representation to new agents is still constrained by the training process.
> >
> > **Q2:** Why is the performance of NegoCollab-FT worse than one-to-one adaptation in the m1m3 and m2m4 settings? Is it due to the negotiation mechanism?
>
>  This result is indeed caused by the negotiation mechanism. As detailed in Section 4.1, the common representation is negotiated between agents m1 and m2 from the initial alliance. **The newly joined agents m3 and m4 do not participate in this negotiation process**, leading to inevitably greater information loss during alignment with the pre-established common representation. In contrast, **the one-to-one adaptation method involves m3 and m4 in training their respective domain adaptation modules**, resulting in relatively smaller information loss. This explains why NegoCollab demonstrates weaker performance than one-to-one adaptation in m1m3 and m2m4 collaboration scenarios. Compared to one-to-one adaptation, **NegoCollab's advantage lies in its lower training cost**: For N agent types involved in collaboration, one-to-one adaptation requires training N² domain adaptation modules, whereas NegoCollab only needs 2N modules.
>
> **The** limitations of NegoCollab when collaborating with new agents can be mitigated by optimizing the composition of agents in the initial **alliance**. In Appendix B.1, we present a comparative analysis of collaborative performance when starting from different initial alliances, as shown in the following table:
>
> |  |      |       |    AP\@0.5   |       |       |  |       |   AP\@0.7    |       |       |
> |------------------|--------|-------|-------|-------|-------|--------|-------|-------|-------|-------|
> | **Initial Alliance** | m1m2   | m3m4  | m1m3  | m2m4  | All   | m1m2   | m3m4  |  m1m3 |  m2m4 | All   |
> | **Protocol**         | **0.792**  | 0.785 | 0.772 | **0.499** | 0.676 | **0.615**  | 0.564 | 0.710 | **0.289** | 0.457 |
> | **m1m3**             | 0.869  | 0.832 | **0.951** | 0.484 | **0.830** | 0.761  | 0.720 | **0.904** | 0.280 | **0.718** |
> | **m1m2**             | **0.872**  | 0.770 | 0.911 | **0.512** | 0.745 | **0.759**  | 0.578 | 0.805 | **0.319** | 0.555 |
> | **m3m4**             | 0.727  | 0.840 | **0.914** | 0.506 | **0.737** | 0.550  | 0.726 | **0.840** | 0.289 | **0.562** |
>
> Table 2: Performance comparison when negotiating common representations from different initial alliances. The "Initial Alliance" column indicates the agents in the initial alliance, while the remaining agents are new agents.
>
>  Based on the comparisons of line 1 and line 3, as well as line 2 and line 4, a qualitative conclusion can be drawn: forming the initial alliance with agents possessing optimally performing perception encoders and including more diverse agent types enhances the generalization capability of the common representation, thereby improving its adaptability to collaboration with new agents.
>
> &nbsp;
>
>  > **Q3:** What is the coordinate system of the BEV common feature map? Does it depend on the relative positions of the collaborating vehicles?
>
>  **During training, both the common representation and each modality's local representations share the same coordinate system**, representing environmental information from identical perspectives, **independent of the relative position of the vehicle**. During the training phase when loading data, NegoCollab uniformly batches perception data from all perspectives in the scene, duplicates it N times, and feeds identical input to N different modality's encoders to obtain paired features of each modality, and subsequently calculating the loss and generating the common representation.
>
> **During inference**, the data loading process distinguishes between different perspectives. The **BEV common features** are transformed from each vehicle's local features, **maintaining coordinate systems aligned with their respective ego perspectives and incorporating relative vehicle positions**.
>
> We will refine the description in lines 158-162 in the revised version to improve reader comprehension.
>
> &nbsp;
>
> > **W:**  The proposed approach is only evaluated on a simulation dataset, whereas the heterogeneous problem in real-world scenarios is more challenging.
>
> We evaluated and compared the performance of NegoCollab against baseline methods on the real-world dataset V2V4Real [1], as shown in the following table:
>
> |                   | Methods    | AP\@0.5 | AP\@0.7 |
> |-------------------|-----------|--------|--------|
> | One to one Adaptation  | MPDA      | 0.613  | 0.400  |
> |                       | PnPDA     | 0.598  | 0.385  |
> |-------------------|-----------|--------|--------|
> |   Align to Common    | MPDA-P    | 0.467  | 0.334  |
> |                | PnPDA-P   | 0.435  | 0.323  |
> |                   | STAMP     | 0.466  | 0.345  |
> |                   | NegoCollab| **0.605**  | **0.397**  |
>
> Table 3: Performance comparison of heterogeneous collaboration on V2V4Real. Agents involved in the collaboration are m1 and m3.
>
> As shown, NegoCollab achieves outstanding collaborative performance even in real-world environments. These experimental results will be included in the revised version.
>
> &nbsp;
>
> [1] V2V4Real: A Real-world Large-scale Dataset for Vehicle-to-Vehicle Cooperative Perception. CVPR2023

---

> > ### Comment · Reviewer_fF95 · 2025-08-04
> >
> > Thank you for your reply. The experimental results address my concerns to a large extent. I have carefully read the authors' response as well as the discussion with other reviewers. I have decided to keep my relatively positive score.

---

> > > ### Author Response · Authors · 2025-08-06
> > > **Thank you very much for your feedback!**
> > >
> > > Thank you very much for your feedback. We will address all the limitations you identified in the revised version, and will supplement additional experiments involving late fusion and real-world datasets.

---

### Official Review · Reviewer_s7iD · 2025-07-01

**Clarity:** 3
**Significance:** 1
**Originality:** 3
**Rating:** 4
**Confidence:** 4

**Summary:**

The authors present NegoCollab, a framework for collaborative perception among agents with different sensors and models. The key idea is to "negotiate" a common representation space that all agents can map to, rather than forcing everyone to conform to one specific agent's representation. This is done using a "negotiator" module trained with a complex, multi-part loss function involving distribution, structural, and pragmatic alignment. The authors show strong performance on the OPV2V-H dataset.

**Questions:**

1. The same as point 4 in the Weaknesses. How is this different from a fusion network?
2. The loss function is complex. The "pragmatic alignment" part, which needs a whole separate network, seems like a lot of extra work. Have you tried removing it? I'm curious how much of your performance gain is due to this versus the core idea.
3. The paper's biggest blind spot is scalability. What's the plan when a new agent type, one that wasn't part of the initial "negotiation," wants to join the collaboration? Do you have to retrain everything from scratch? If so, the "low training cost" claim from the abstract seems misleading, as the real cost is in re-integration.  If you can clarify this, I would be pleased to improve the rating.

**Ethical Concerns:**

["NO or VERY MINOR ethics concerns only"]

**Final Justification:**

In light of the fact that the authors have addressed some of my concerns, I have increased my score.

**Limitations:**

The primary limitation of this work is the challenge of dynamically integrating new agents into the system.

**Paper Formatting Concerns:**

Looks good overall

**Quality:**

2

**Strengths And Weaknesses:**

Strengths:
1. For a fixed set of heterogeneous agents, the proposed "negotiation" process and the complex alignment losses do achieve excellent performance.
2. The idea that a democratically "negotiated" space might be better than a "dictated" one is appealing at first glance.

Weaknesses:
1. This paper tackles a "closed-world" problem where the set of agent types is predefined. This is a significant step back from the "open-world" formulation presented in recent work like HEAL (Lu et al., ICLR 2024), which explicitly addresses the critical challenge of continuously integrating new, unseen agent types. NegoCollab's approach, by design, ignores the most crucial aspect of real-world deployment: scalability and extensibility.
2. The core design of NegoCollab is inherently non-extensible. When a new agent needs to join, the entire "negotiated" space becomes suboptimal. This forces a choice between two bad options: either the new agent suffers from poor alignment, or the entire collective must be retrained at a massive cost. This contrasts sharply with HEAL's "backward alignment," which provides a clear, low-cost path for new agents to join.
3.  The paper introduces a highly complex loss function with distribution, structural, and even a "pragmatic" alignment component , kind of over-engineered.
4. The core concept is "negotiation," which implies a dynamic, give-and-take process. But your negotiator is trained once and then fixed. How is this different from a multi-input fusion network? The term feels like a bit of a stretch.

---

> ### Author Rebuttal · Authors · 2025-07-27
>
> Thank you very much for dedicating your time and expertise to review our submission. The depth of your analysis and the constructive criticism you have provided are immensely beneficial. I sincerely hope my reply can answer your questions.
>
> > **W4:** The core concept is "negotiation," which implies a dynamic, give-and-take process. But your negotiator is trained once and then fixed. How is this different from a multi-input fusion network? The term feels like a bit of a stretch.
> >
> > **Q1:** The same as point 4 in the Weaknesses. How is this different from a fusion network?
>
> Architecturally, the **Negotiator** does resemble **multi-input fusion networks** in structure. However, **their functional roles within the collaborative framework differ fundamentally**, as we elaborate below:
>
> - **Input/Output**: The **negotiator** takes local representations encoded by each modality's encoders from the **same perspective** of environmental information as **input**, and **outputs a common representation**. In contrast, the **input** of the **multi-input fusion network** is the features obtained by agents from encoding the environmental information from **different perspectives**, and **outputs features fused with multi-perspective information**.
>
> - **Estimator:** The estimator in the Negotiator **evaluates each modality's contribution to the shared representation**, whereas the foreground estimator in multi-input fusion networks **assesses the probability of foreground objects** appearing at corresponding locations in BEV.
>
> - **Role of Module:** The Negotiator's role is to **negotiate the common representations** while implicitly preserving their information through model parameters, whereas multi-input fusion networks are designed to **integrate multi-view information**.
>
> In NegoCollab, **the concept of "negotiation" is embodied in the mutual compromise among agents within the initial alliance** when negotiating the common representation from their respective local representations. For dynamically joined new agents participating in collaboration, they only need to align with the common representation negotiated by the initial alliance.
>
> &nbsp;
>
> > **W3:** The paper introduces a highly complex loss function with distribution, structural, and even a "pragmatic" alignment component, kind of over-engineered.
> >
> > **Q2:** The loss function is complex. The "pragmatic alignment" part, which needs a whole separate network, seems like a lot of extra work. Have you tried removing it? I'm curious how much of your performance gain is due to this versus the core idea.
>
> The multi-dimensional alignment loss plays a critical role in facilitating better alignment between unimodal local representations and the multimodal common representation. In Section 4.3, we perform comprehensive ablation studies on the individual components of the multi-dimensional alignment loss. The comparative results after completing the first training phase are summarized in the table below:
>
> | uni-dis | uni-stru | uni-pragma | AP\@0.5 | Inc.   | AP\@0.7 | Inc.    |
> |:---------:|:----------:|:------------:|:--------:|:--------:|:--------:|:---------:|
> | ✓       |          |            | 0.609  | -      | 0.496  | -       |
> | ✓       | ✓        |            | 0.655  | +7%    | 0.532  | +6.8%   |
> | ✓       |          | ✓          | 0.671  | +9.2%  | 0.538  | +7.8%   |
> | ✓       | ✓        | ✓          | **0.711** | **+14.3%** | **0.566** | **+12.4%** |
>
> Table 1. Ablation study of the multi-dimensional alignment loss. The collaborating agents are m1 and m2.
>
> Taking the performance under AP@0.5 as an example: when solely adding the structural **consistency loss, performance improves by 7%**; when only incorporating the **pragmatic consistency loss, performance gains approximately 10%**; when **both consistency losses** are included, performance **increases by about 14%**. These results clearly demonstrate the significance of multi-dimensional alignment loss for enhancing heterogeneous collaboration performance. Furthermore, the **separate network required for auxiliary pragmatic alignment** is a 2D occupancy detection head, implemented as a single-layer 2D convolutional network **with approximately 1.3K parameters, introducing virtually no additional training overhead**.
>
> The design of the multi-dimensional alignment loss stems from this **analysis: The essence of aligning local representations to the common representation lies in transforming single-modal features into multi-modal features, which cannot be adequately achieved solely through conventional distribution consistency loss that merely constrains feature distributions.** This is because the common representation is negotiated from multiple different types of agents and contains multi-modal information, while the local representation is encoded by the local encoder from observed data and contains only single-modal information. Taking the common representation negotiation between m1 and m2 as an example: m1 and m2 employ LiDAR and camera sensors respectively, with their local representations containing only single-modal information from the corresponding sensors. The common representation negotiated from m1 and m2 combines camera-LiDAR fused modalities. However, **inherent differences** exist between camera and LiDAR modalities **in terms of data format, spatial structure, and pragmatic information**. These differences are consequently reflected in the disparities between single-modal (camera/LiDAR) features and fused-modal features. Therefore, **aligning such cross-modal features requires multi-dimensional consistency constraints encompassing distribution, structural relationships, and pragmatic information between features.**
>
> In the revised version, we will refine the descriptions of the multi-dimensional alignment loss and its ablation studies (Lines 189-195 and 293-301) to better clarify their functions.
>
> &nbsp;
>
> >**W1:** This paper tackles a "closed-world" problem where the set of agent types is predefined.
> >
> > **W2:** The core design of NegoCollab is inherently non-extensible.
> >
> > **Q3:** The paper's biggest blind spot is scalability. What's the plan when a new agent type, one that wasn't part of the initial "negotiation," wants to join the collaboration? Do you have to retrain everything from scratch? If so, the "low training cost" claim from the abstract seems misleading, as the real cost is in re-integration. If you can clarify this, I would be pleased to improve the rating.
>
> For dynamically joining **new agents**, collaborative participation can be achieved by training a pair of **plug-and-play sender-receiver** to align with the negotiated common representation. The training process **only requires parameter updates for the new agent's sender and receiver modules while eliminating the need to retrain the negotiator network**, thus providing **an efficient solution for heterogeneous collaboration in "open-world" scenarios with minimal training cost**.
>
> We explained the training process when a new agent joins in Section 4.1. Specifically, in the first stage, **the common representation is generated from the negotiator and the encoder of the agents in initial alliance.** The cyclic consistency loss and multi-dimensional alignment loss are calculated in the same way as described in Section 3.2.2 and Section 3.2.3. In the second stage, the collaborative task loss is calculated as the collaborative detection loss between the new agent and the agents in the initial alliance. We will add a diagram of the training process for a new agent in the appendix of the revised version.
>
> Notably, since new agents do not participate in the negotiation process, aligning them to the common representation inevitably incurs greater information loss. This indeed represents a limitation of NegoCollab in "open-world" collaboration scenarios. To mitigate this limitation, we investigate NegoCollab's collaborative performance when the common representation is negotiated from different initial alliances. Our analysis concludes that selecting initial alliance agents with optimally performing perception encoders and including more diverse agent types in the initial alliance can enhance the generalization capability of the common representation, thereby improving collaborative performance in "open-world" settings. The relevant results are presented in Appendix B.1, with selected findings shown below:
>
> |  |      |       |    AP\@0.5   |       |       |  |       |   AP\@0.7    |       |       |
> |------------------|--------|-------|-------|-------|-------|--------|-------|-------|-------|-------|
> | **Initial Alliance**  | m1m2   | m3m4  | m1m3  | m2m4  | All   | m1m2   | m3m4  |  m1m3 |  m2m4 | All   |
> | **Protocol**          | 0.792  | 0.785 | 0.772 | 0.499 | 0.676 | 0.615  | 0.564 | 0.710 | 0.289 | 0.457 |
> | **m1m3**              | 0.869  | 0.832 | 0.951 | 0.484 | 0.830 | 0.761  | 0.720 | 0.904 | 0.280 | 0.718 |
> | **m1m2**              | 0.872  | 0.770 | 0.911 | 0.512 | 0.745 | 0.759  | 0.578 | 0.805 | 0.319 | 0.555 |
> | **m3m4**              | 0.727  | 0.840 | 0.914 | 0.506 | 0.737 | 0.550  | 0.726 | 0.840 | 0.289 | 0.562 |
>
> Table 2: Performance comparison when negotiating common representations from different initial alliances.

---

### Official Review · Reviewer_py3y · 2025-07-02

**Clarity:** 2
**Significance:** 3
**Originality:** 3
**Rating:** 4
**Confidence:** 3

**Summary:**

This paper introduces NegoCollab, a framework for collaborative perception among heterogeneous agents, particularly in autonomous driving scenarios. The key idea is to negotiate a common representation space by aggregating local representations from multiple modalities (e.g., LiDAR, camera) through a dedicated negotiator during training, thus minimizing domain gaps between agents with differing sensors and backbones. Training is supervised by distribution, structure, and pragmatic alignment losses to ensure comprehensive and meaningful alignment. The experimental results on the OPV2V-H dataset show that NegoCollab achieves superior performance compared to existing one-to-one and common representation-based approaches, demonstrating improved adaptability and knowledge transfer across heterogeneous agent types.

**Questions:**

See Weekness.

**Ethical Concerns:**

["NO or VERY MINOR ethics concerns only"]

**Final Justification:**

Overall, it is a good paper. The author have solved my main concerns. It's glad to see the author's contribution, I prefer to accept this paper.

**Limitations:**

Yes

**Quality:**

3

**Strengths And Weaknesses:**

*Strengths*
1. The paper addresses a relevant and practical challenge in collaborative perception: handling immutable heterogeneity among agents by introducing a negotiator for common representation. This is a meaningful contribution to autonomy and multi-agent systems.
2. The methodology is well constructed, combining a plug-and-play sender/receiver architecture with a negotiator for representation fusion. The distinction from solely protocol-agent-based approaches is well articulated both in the Introduction and Figure 1.


*Weakness*

1.Lack of Architectural Component Ablation.

The paper introduces several novel architectural components, such as the ConvNeXt-based Recombiner, the Sparse Transformer-based Aligner/Converter, and the FPN structure within the Negotiator.  However, the ablation studies do not sufficiently justify these specific design choices. It remains unclear how much performance gain is attributable to these sophisticated modules compared to simpler alternatives (e.g., using MLPs for recombination or a non-pyramidal structure for the negotiator). A more thorough ablation on these architectural sub-components would be necessary to validate their effectiveness and contribution to the overall framework.

2. The Role of the "Local Prompt" is Not Experimentally Validated.

A key design element of the Receiver is the use of a "local prompt"—the output of the sender's recombiner as the query vector for the converter. This is proposed to provide local modality guidance during the feature transformation process.  However, the importance of this mechanism is not ablated

---

> ### Author Rebuttal · Authors · 2025-07-28
>
> We sincerely appreciate your recognition of our work's contributions to multi-agent system development. Your constructive suggestions are invaluable to refining this work. In the following, we will address your concerns regarding the weaknesses and show additional experimental results.
>
> >  **W1:** Lack of Architectural Component Ablation.
>
> We respectively validated the performance of NegoCollab when setting Recombiner, Aligner, and Negotiator to different structures (while keeping other modules unchanged when modifying one component's structure), as shown in the table below:
>
> | Components   | Settings         | AP\@0.5 | AP\@0.7 | #Params# |
> |--------------|------------------|--------|--------|----------|
> | Recombiner   | ResMlp           | 0.633  | 0.510  | 0.1 M    |
> |              | FANet            | 0.649  | 0.492  | 1.7 M    |
> |              | Conevxt           | **0.711**  | **0.566**  | 0.3 M    |
> |              | Sparse Transformer | 0.596  | 0.487  | 0.2 M    |
> | Aligner      | ResMlp           | 0.697  | 0.527  | 0.1 M    |
> |              | FANet            | 0.696  | 0.563  | 1.7 M    |
> |              | Convext           | 0.702  | 0.542  | 0.3 M    |
> |              | Sparse Transformer | **0.711**  | **0.566**  | 0.2 M    |
> | Negotiator   | ResMlp           | 0.705  | 0.565  | 1.8 M    |
> |              | Convext           | 0.706  | 0.566  | 2.7 M    |
> |              | Sparse Transformer | 0.706  | 0.564  | 2.1 M    |
> |              | FPN              | **0.711**  | **0.566**  | 1.2 M    |
>
> Table 1: Architectural Component Ablation.
>
>
> In the table, **#Params#** represents the size of parameter of module when it uses different settings. The evaluation involves collaborating agents m1 and m2, showing their collaborative performance prior to downstream task adaptation training. The "ResMLP" is a network with a multilayer perceptron (MLP) backbone, FANet [1] employs a 2D convolutional backbone in an encoder-decoder architecture for feature space adaptation.
>
> The comparative results demonstrate that **NegoCollab achieves optimal performance when the Recombiner and Aligner are configured as Convext and Sparse Transformer respectively in their corresponding experimental groups**. This superior performance arises from our two-stage feature transformation pipeline: local detail refinement via the Recombiner and global representation style transfer through the Aligner. **The architectural properties of Convext and Sparse Transformer make them ideally suited for local refinement and representation transformation tasks respectively**, allowing each module in NegoCollab to operate at maximum effectiveness. **For the Negotiator component, our implemented FPN structure attains peak performance with minimal parameter requirements.** The above results fully demonstrate the rationality of each component's design in NegoCollab.
>
> [1] Feature-Align Network with Knowledge Distillation for Efficient Denoising. WACV2022
>
> &nbsp;
>
> >  **W2:** The Role of the "Local Prompt" is Not Experimentally Validated.
>
> We evaluated NegoCollab's performance with and without Local Prompt integration, as shown in the following table:
>
> | Local Prompt | AP\@0.5 | AP\@0.7 |
> |--------------|--------|--------|
> | w/o          | 0.672  | 0.547  |
> | w            | 0.711  | 0.566  |
>
> Table 2: Component ablation of local prompt. The collaborating agents involved were m1 and m2, with the table presenting the collaborative performance before downstream task adaptation training.
>
> The results demonstrate that integrating **local prompt improves NegoCollab's performance by 5.8% in AP@0.5 and 3.5% in AP@0.7**, confirming the effectiveness of this module.
>
> &nbsp;
>
>  Finally, we sincerely appreciate your constructive feedback on our work, and we will incorporate all the aforementioned ablation studies in the revised version.

---

> ### Comment · Reviewer_py3y · 2025-08-07
>
> Thank you for the detailed responses to the related questions, which truly solved my concerns. Overall, it is a good paper. I will update my final justification and also appreciate the author's contributions.

---

### Official Review · Reviewer_K5st · 2025-07-03

**Clarity:** 3
**Significance:** 3
**Originality:** 3
**Rating:** 4
**Confidence:** 3

**Summary:**

This paper addresses the challenge of immutable heterogeneity in collaborative perception—where agents with different, fixed perception models share intermediate features that suffer from domain gaps—by introducing NegoCollab, a framework that negotiates a common representation from each agent’s local representations via a feature-pyramid-based negotiator, and then employs plug-and-play sender–receiver modules to bidirectionally map between local and common feature spaces. To supervise this process, the authors propose a multi-dimensional alignment loss comprising distribution alignment, structural alignment (enforcing consistency of scene-component relations), and pragmatic alignment (via a unified 2D occupancy prediction task)

**Questions:**

N/A

**Ethical Concerns:**

["NO or VERY MINOR ethics concerns only"]

**Final Justification:**

Thanks for the rebuttal. I keep my score Borderline accept

**Limitations:**

yes

**Quality:**

3

**Strengths And Weaknesses:**

Strengths:

1. The negotiator generates a common representation by weighting multi-level features from each modality, substantially reducing domain gaps and easing sender–receiver alignment

2. Combining distribution, structural, and pragmatic alignment fosters richer knowledge distillation from the common space, improving the expressiveness of local senders and boosting collaborative performance.

Weakness:

1. Once negotiated, the shared representation is static, leading to inevitable information loss and limited adaptability when integrating entirely new agent types.

2. All experiments are conducted on the synthetic OPV2V-H dataset; real-world validation on diverse traffic scenes is absent, leaving open questions about generalization in more complex environments.

3. While domain-gap reduction is well addressed, the approach does not consider bandwidth limitations, latency, or security/privacy aspects essential for real-world collaborative perception deployments.

---

> ### Author Rebuttal · Authors · 2025-07-28
>
> To the esteemed reviewer, I deeply appreciate the time and effort you've put into reviewing my work. Your suggestions are very helpful and will be carefully considered. Now let me address some concerns for you.
>
> > **W1:** Once negotiated, the shared representation is static, leading to inevitable information loss and limited adaptability when integrating entirely new agent types.
>
> Since the negotiation of common representations involves only agents within the initial alliance, it inevitably causes information loss for new agents. **Optimizing the composition of the initial alliance to enhance the generalization capability of common representations can partially mitigate this issue**. We provide an experimental analysis of collaborative performance when starting from different alliances in Appendix B.1, as shown in the following table:
>
> |  |      |       |    AP\@0.5   |       |       |  |       |   AP\@0.7    |       |       |
> |------------------|--------|-------|-------|-------|-------|--------|-------|-------|-------|-------|
> | **Initial Alliance** | m1m2   | m3m4  | m1m3  | m2m4  | All   | m1m2   | m3m4  |  m1m3 |  m2m4 | All   |
> | **Protocol**         | **0.792**  | 0.785 | 0.772 | **0.499** | 0.676 | **0.615**  | 0.564 | 0.710 | **0.289** | 0.457 |
> | **m1m3**             | 0.869  | 0.832 | **0.951** | 0.484 | **0.830** | 0.761  | 0.720 | **0.904** | 0.280 | **0.718** |
> | **m1m2**             | **0.872**  | 0.770 | 0.911 | **0.512** | 0.745 | **0.759**  | 0.578 | 0.805 | **0.319** | 0.555 |
> | **m3m4**             | 0.727  | 0.840 | **0.914** | 0.506 | **0.737** | 0.550  | 0.726 | **0.840** | 0.289 | **0.562** |
>
> Table 1: Performance comparison when negotiating common representations from different initial alliances. The "Initial Alliance" column indicates the agents in the initial alliance, while the remaining agents are new agents.
>
> The comparison between line 1 ("Protocol", initial alliance containing only LiDAR-equipped agents) and line 3 ("m1m2", initial allliance containing both LiDAR and camera-equipped agents) demonstrates that common representations negotiated from the initial alliance composed with more types of agents achieve better performance in most collaborative scenarios. The comparison between line 2 ("m1m3") and line 3 ("m3m4") further shows that common representations negotiated from the initial alliance composed of relatively better-performing agents, m1 and m3, yield superior results in most cases. These findings support the **qualitative conclusion** that **selecting agents with better perception encoders for the initial alliance and including more agent types can enhance the generalization capability of common representations**, thereby improving collaboration with new agents.
>
> &nbsp;
>
> > **W2:** All experiments are conducted on the synthetic OPV2V-H dataset; real-world validation on diverse traffic scenes is absent, leaving open questions about generalization in more complex environments.
>
> We evaluated and compared the performance of NegoCollab against baseline methods on the real-world dataset V2V4Real [1], as shown in the following table:
>
> |                   | Methods    | AP\@0.5 | AP\@0.7 |
> |-------------------|-----------|--------|--------|
> | One to one Adaptation  | MPDA      | 0.613  | 0.400  |
> |                       | PnPDA     | 0.598  | 0.385  |
> |-------------------|-----------|--------|--------|
> |   Align to Common    | MPDA-P    | 0.467  | 0.334  |
> |                | PnPDA-P   | 0.435  | 0.323  |
> |                   | STAMP     | 0.466  | 0.345  |
> |                   | NegoCollab| **0.605**  | **0.397**    |
>
> Table 3: Performance comparison of heterogeneous collaboration on V2V4Real. Agents involved in the collaboration are m1 and m3.
>
> As shown, NegoCollab achieves outstanding collaborative performance even in real-world environments. These experimental results will be included in the revised version.
>
> [1] V2V4Real: A Real-world Large-scale Dataset for Vehicle-to-Vehicle Cooperative Perception. CVPR2023
>
> &nbsp;
>
> > **W3:** While domain-gap reduction is well addressed, the approach does not consider bandwidth limitations, latency, or security/privacy aspects essential for real-world collaborative perception deployments.
>
> NegoCollab eliminates the domain gap by introducing a pair of sender and receiver modules, which serve as additional plug-and-play components to the original homogeneous collaboration model of the agent. Aspects essential for real-world collaborative perception deployments. such as bandwidth limitations, latency, and security/privacy are **handled by the original homogeneous collaboration model**. As long as the original homogeneous collaboration model incorporates additional designs addressing these challenges, NegoCollab can readily adapt to real-world environments.
>
> To verify this, we simulated the **positioning errors** existing in the real world in the synthetic dataset OPV2V-H, and evaluated NegoCollab's performance under these conditions (see Figure 3 in the main text). The results show that NegoCollab maintains strong robustness even with increasing localization errors, **demonstrating its preliminary adaptability to complex real-world environments**.

---

### Note · Authors · 2025-08-11

NegoCollab addresses the challenge of immutable heterogeneity in collaborative perception by proposing a method that generates common representation through negotiation. This approach significantly reduces the domain gap between the common representation and individual agents' local representations, thereby enhancing heterogeneous collaborative performance. All 4 reviewers praised the novelty and rationality of the "generate common representation through negotiation" method in their reviews, acknowledging its contribution to multi-agent system development.

Regarding limitations, the reviewer's concerns primarily focus on applicable conditions, architectural details, and performance in broader environments. We addressed all the reviewers' questions point by point, with necessary additional experiments included in the rebuttal. Here are several key concerns raised by the reviewers and our corresponding responses:

>**1. How to collaborate with new agents and whether can solve the "open-world" problem?** (@s7iD-W1,W2,Q3)

Reviewer @s7iD argued in W1 that NegoCollab addresses a "closed-world" problem, and noted in W2 and Q2 that NegoCollab is non-extensible and cannot effectively collaborate with new agents. This misunderstanding may stem from overlooking our setting of the initial alliance (m1, m2) and new agents (m3, m4) in Section 4.1, as well as the description of the "New Agent Alignment" method.

In our rebuttal to reviewer @s7iD, we provided a detailed explanation of the training process for new agents joining the alliance. Our analysis demonstrates that new agents achieve collaboration by aligning with existing common representation, requiring only minimal additional training of a pair of sender-receiver. This approach can address the "open-world" problem with minimal training overhead. The performance when collaborating with new agents is presented in Table 1 of the main paper.

>**2. Inevitably information loss when new agents join.** (@K5st-W1, @fF95-W, Q2)

Both reviewers @K5st and @fF95 pointed out that NegoCollab inevitably incurs information loss when new agents join the alliance. We propose that this problem can be alleviated by optimizing the composition of the agents within the initial alliance to enhance the generalization ability of the common representation. Through experiments, we derived two qualitative guidelines for initial alliance optimization. The experiment and analysis have been added to rebuttal responses to @K5st and @fF95.

---

### Decision · Program_Chairs · 2025-09-17

**Decision:**

Accept (poster)

**Comment:**

The paper proposes NegoCollab, a novel framework for heterogeneous collaborative perception that negotiates a common representation across multiple agents and leverages sender–receiver mappings together with multi-dimensional alignment losses.

Reviewers agree that the idea of negotiated representation is both original and meaningful for reducing domain gaps in multi-agent perception. The methodology is technically sound, combining distributional, structural, and pragmatic alignment in a modular design. Empirical results on OPV2V-H show consistent improvements over strong baselines, and the rebuttal further added results on V2V4Real, strengthening confidence in the approach. Other highlighted strengths include clear motivation, strong architectural design, and the ability to generalize to different agent types without retraining entire models.

At the same time, some limitations remain. The negotiated representation is essentially static and scalability to unseen agents is not fully explored. Evaluation is still simulation-heavy, and the complexity of the alignment loss design may pose implementation challenges.

Nevertheless, these issues do not undermine the core contributions, which are both technically sound and practically relevant. We recommend acceptance, with the following suggestions for the camera-ready: (1) providing deeper ablations on the role of each alignment loss, (2) clarifying scalability to dynamic and real-world heterogeneous agents, and (3) including the experiments added during the rebuttal.